# Learning Causes of Functional Dynamic Targets: Screening and Local Methods

**DOI:** 10.3390/e26070541

**Published:** 2024-06-24

**Authors:** Ruiqi Zhao, Xiaoxia Yang, Yangbo He

**Affiliations:** 1School of Mathematical Sciences, Peking University, Beijing 100871, China; zhaorq@pku.edu.cn; 2College of Science, Beijing Forestry University, Beijing 100083, China

**Keywords:** screening method, local structure learning, functional dynamic target, direct causes, indirect causes

## Abstract

This paper addresses the challenge of identifying causes for functional dynamic targets, which are functions of various variables over time. We develop screening and local learning methods to learn the direct causes of the target, as well as all indirect causes up to a given distance. We first discuss the modeling of the functional dynamic target. Then, we propose a screening method to select the variables that are significantly correlated with the target. On this basis, we introduce an algorithm that combines screening and structural learning techniques to uncover the causal structure among the target and its causes. To tackle the distance effect, where long causal paths weaken correlation, we propose a local method to discover the direct causes of the target in these significant variables and further sequentially find all indirect causes up to a given distance. We show theoretically that our proposed methods can learn the causes correctly under some regular assumptions. Experiments based on synthetic data also show that the proposed methods perform well in learning the causes of the target.

## 1. Introduction

Identifying the causes of a target variable is a primary objective in numerous research studies. Sometimes, these target variables are dynamic, observed at distinct time intervals, and typically characterized by functions or distinct curves that depend on other variables and time. We call them functional dynamic targets. For example, in nature, the growth of animals and plants is usually multistage and nonlinear with respect to time [1,2,3]. The popular growth curve functions, including Logistic, Gompertz, Richards, Hossfeld IV, and Double-Logistic functions, have *S* shapes [3], and have been widely used to model the patterns of growth. In psychological and cognitive science, researchers usually fit individual learning and forgetting curves by power functions; individuals may have different curve parameters [4,5].

The causal graphical model is widely used for the automated derivation of causal influences in variables [6,7,8,9] and demonstrates excellent performance in presenting complex causal relationships between multiple variables and expressing causal hypotheses [7,10,11]. In this paper, we aim to identify the underlying causes of these functional dynamic targets using the graphical model. There are three main challenges for this purpose. Firstly, identifying the causes is generally more challenging than exploring associations, even though the latter has received substantial attention, as evidenced by the extensive use of Genome-Wide Association Studies (GWAS) within the field of bioinformatics. Secondly, it is difficult to use a causal graphical model to represent the generating mechanism of dynamic targets and to find the causes of the targets from observational data when the number of variables is very large. For example, one needs to find the genes that affect the growth curve of individuals from more than thousands of Single-Nucleotide Polymorphisms (SNPs). Finally, the variables considered are mixed, which increases the complexity of representing and learning the causal model. We discuss these three challenges in detail below.

First of all, traditional statistical methods can only discover correlations between variables, rather than causal relationships, which may give false positive or false negative results for finding the real causes of the target. In fact, the association between the target and variables may originate from different causal mechanisms. For example, Figure 1 displays several different causal mechanisms possibly resulting in a statistically significant association between the target and variables. In Figure 1, $X_{1},X_{2},X_{3}$ are three random variables, $Y=(Y_{1},\cdots ,Y_{n})$ is a vector representing a functional dynamic target, in which $Y_{i},i=1,\ldots ,n$ are states of the target in *n* time points, and direct edges represent direct causal relations among them. Using the statistical methods, we are very likely to find that $X_{1}$ is associated with Y significantly in all four cases. However, it is hard to identify whether $X_{1}$ is a real cause of Y without further causal learning. As shown in Figure 1, $X_{1}$ might be a direct cause of Y in Figure 1a,c, a cause but not a direct cause in Figure 1b, and not a cause in Figure 1d.

In addition, when the number of candidate variables is very huge, both learning causal structures and discovering target causes become very difficult. In fact, learning the complete causal graph is redundant and wasteful for the task of finding causes, as the focus should be on the target variable’s local structure. PCD-by-PCD algorithm [12] is adept at identifying such local structures and efficiently distinguishing parents, children, and some descendants. The MB-by-MB method [13], in contrast, simplifies this by learning Markov Blanket (MB) sets for identifying direct causes/effects, leveraging simpler and quicker techniques compared with PCD sets with methods like PCMB, STMB, and EEMB [14,15,16]. The CMB algorithm further streamlines this process using a topology-based MB discovery approach [17]. However, Ling [18] pointed out that Expand-Backtracking-type algorithms, such as the PCD-by-PCD and CMB algorithms, may overlook some v-structures, leading to numerous incorrect edge orientations. To tackle these issues, the APSL algorithm was introduced and designed to learn the subgraph within a specific distance centered around the target variable. Nonetheless, its dependence on the FCBF method for Markov Blanket learning tends to produce approximate sets rather than precise ones [19]. Furthermore, Ling [18] emphasized learning the local graph within a certain distance from the target rather than focusing on the causes of the target.

Finally, the variables in question are varied; specifically, the targets consist of dynamic time series or complex curves, while the other variables may be either discrete or continuous. Consequently, measuring the connections between the target and other variables presents significant challenges. For instance, traditional statistical methods used to assess independence or conditional independence between variables and complex targets might not only be inefficient but also ineffective, especially when there is an insufficient sample size to accurately measure high-order conditional independence.

In this paper, we introduce a causal graphical model tailored for analyzing dynamic targets and propose two methods to identify the causes of such a functional dynamic target assuming no hidden variables or selection biases. Initially, after establishing our dynamic target causal graphical model, we conduct an association analysis to filter out most variables unrelated to the target. With data from the remaining significantly associated variables, we then combine the screening method with structural learning algorithms and introduce the SSL algorithm to identify the causes of the target. Finally, to mitigate the distance effects that can mask the association between a cause and the target in data sets where the causal chain from cause to target is excessively long, we propose a local method. This method initially identifies the direct causes of the target and then proceeds to learn the causes sequentially in reverse order along the causal path.

The main contributions of this paper include the following:We introduce a causal graphical model that combines Bayesian networks and functional dynamic targets to represent the causal mechanism of variables and the target.We present a screening method that significantly reduces the dimensions of potential factors and combines it with structural learning algorithms to learn the causes of a given target and prove that all identifiable causes can be learned correctly.We propose a screening-based and local method to learn the causes of the functional dynamic target up to any given distance among all factors. This method is helpful when association disappears due to the long distance between indirect causes and the target.We experimentally study our proposed method on a simulation data set to demonstrate the validity of the proposed methods.

## 2. Preliminary

Before introducing the main results of this paper, we need to clarify some definitions and notations related to graphs. Furthermore, unless otherwise specified, we use capital letters such as *V* to denote variables or vertices, boldface letters such as V to denote variable sets or vectors, and lowercase letters such as *v* and v to denote the realization of a variable or vector, respectively.

A graph G is a pair (V,E), in which $V=\{V_{1},\cdots ,V_{p}\}$ is the vertex set and $E\subseteq E^{*}\left(V\right):=\left(V\times V\right)\setminus \left\{\left(V_{i},V_{i}\right)|V_{i}\in V\right\}$ is the edge set. To simplify the symbols, we use V to represent both random variables and the corresponding nodes in the graph. For any two nodes $V_{i},V_{j}\in V$, an undirected edge between $V_{i}$ and $V_{j}$, denoted by $V_{i}-V_{j}$, is an edge satisfying $(V_{i},V_{j})\in E$ and $(V_{j},V_{i})\in E$, while a directed edge between $V_{i}$ and $V_{j}$, denoted by $V_{i}\to V_{j}$, is an edge satisfying $(V_{i},V_{j})\in E$ and $(V_{j},V_{i})\notin E$. If all edges in a graph are undirected (directed), the graph is called an undirected (directed) graph. If a graph has both undirected and directed edges, then it is called a partially directed graph. For a given graph G, we use V(G) and E(G) to denote its vertex set and edge set, respectively, where G can be an undirected, directed, or partially directed graph. For any $V^{\prime }\subseteq V$, the induced subgraph of G over $V^{\prime }$, denoted by $G(V^{\prime })$ or $G_{V^{\prime }}$, is the graph with vertex set $V^{\prime }$ and edge set $E(V^{\prime })\subseteq E$ containing all and only edges between vertices in $V^{\prime }$, that is, $G_{V^{\prime }}=\left(V^{\prime },E\left(V^{\prime }\right)\right)$, where $E\left(V^{\prime }\right):=E\cap \left(V^{\prime }\times V^{\prime }\right)$.

In a graph G, $V_{i}$ is a parent of $V_{j}$ and $V_{j}$ is a child of $V_{i}$ if the directed edge $V_{i}\to V_{j}$ is in G. $V_{i}$ and $V_{j}$ are neighbors of each other if the undirected edge $V_{i}-V_{j}$ is in G. $V_{i}$ and $V_{j}$ are called adjacent if they are connected by an edge, regardless of whether the edge is directed or undirected. We use $Pa\left(V_{i},G\right),Ch\left(V_{i},G\right),Ne\left(V_{i},G\right),Adj\left(V_{i},G\right)$ to denote the sets of parents, children, neighbors, and adjacent vertices of $V_{i}$ in G, respectively. For any vertex set $V^{\prime }\subseteq V$, the parent set of $V^{\prime }$ in G can be defined as $Pa\left(V^{\prime },G\right)=\cup_{V_{i}\in V^{\prime }}Pa\left(V_{i},G\right)\setminus V^{\prime }$. The sets of children, neighbors, and adjacent vertices of $V^{\prime }$ in G can be defined similarly. A root vertex is the vertex without parents. For any vertex $V_{i}\in V$, the degree of $V_{i}$ in G, denoted by $deg(V_{i},G)$, is the number of $V_{i}$’s adjacent vertices, that is, $deg\left(V_{i},G\right)=\left|Adj\left(V_{i},G\right)\right|$. The skeleton of G, denoted by $G_{s}$, is an undirected graph obtained by transforming all directed edges in G to undirected edges, that is, $G_{s}:=\left(V,E_{S}\right)$, where $E_{s}:=\left\{\left(V_{i},V_{j}\right)\in V\times V|V_{i}\in Adj\left(V_{j},G\right)\right\}$.

The sequence $<V_{i_{1}},\cdots ,V_{i_{n}}>$ in graph G is an ordered collection of distinct vertices $V_{i_{1}},\cdots ,V_{i_{n}}$. A sequence becomes a path, denoted by $\left(V_{i_{1}},\cdots ,V_{i_{n}}\right)$, if every pair of consecutive vertices in the sequence is adjacent in G. The vertices $V_{i_{1}}$ and $V_{i_{n}}$ serve as the endpoints, with the rest being intermediate vertices. For a path $\pi =(V_{i_{1}},\cdots ,V_{i_{n}})$ in G, and for any 1≤k≤n, the subpath from $V_{i_{1}}$ to $V_{i_{k}}$ is $\pi \left(V_{i_{1}},V_{i_{k}}\right)=\left(V_{i_{1}},\cdots ,V_{i_{k}}\right)$, and path π can thus be represented as a combination of its subpaths, denoted by $\pi =\pi \left(V_{i_{1}},V_{i_{k}}\right)⊕\pi \left(V_{i_{k}},V_{i_{n}}\right)$. A path is partially directed if there is no directed edge $V_{i_{k+1}}\to V_{i_{k}}$ in G for any k=1,…,n−1. A partially directed path is directed (or undirected) if all its edges are directed (or undirected). A vertex $V_{i}$ is an ancestor of $V_{j}$ and $V_{j}$ is a descendant of $V_{i}$ if there exists a directed path from $V_{i}$ to $V_{j}$ or $V_{i}=V_{j}$. The sets of ancestors and descendants of $V_{i}$ in the graph G are denoted by $An(V_{i},G)$ and $De(V_{i},G)$, respectively. Furthermore, a vertex $V_{i}$ is a possible ancestor of $V_{j}$ and $V_{j}$ is a possible descendant of $V_{i}$ if there is a partially directed path from $V_{i}$ to $V_{j}$. The sets of possible ancestors and possible descendants of $V_{i}$ in graph G are denoted by $PossAn(V_{i},G)$ and $PossDe(V_{i},G)$, respectively. For any vertex set $V^{\prime }\subseteq V$, the ancestor set of $V^{\prime }$ in graph G is $An\left(V^{\prime },G\right):=\cup_{V_{i}\in V^{\prime }}An\left(V_{i},G\right)$. The sets of possible ancestors and (possible) descendants of $V^{\prime }$ in graph G can be defined similarly.

A (directed, partially directed, or undirected) cycle is a (directed, partially directed, or undirected) path from a node to itself. The length of a path (cycle) is the number of edges on the path (cycle). The distance between two variables $V_{i}$ and $V_{j}$ is the length of the shortest directed path from $V_{i}$ to $V_{j}$. A directed acyclic graph (DAG) is a directed graph without directed cycles, and a partially directed acyclic graph (PDAG) is a partially directed graph without directed cycles. A chain graph is a partially directed graph in which all partially directed cycles are undirected. This indicates that both DAGs and undirected graphs can be considered as specific types of chain graphs.

In a graph G, a v-structure is a tuple $\left(V_{i},V_{j},V_{k}\right)$ satisfying $V_{i}\to V_{j}\leftarrow V_{k}$ with $V_{i}\notin Adj\left(V_{k},G\right)$, in which $V_{j}$ is called a collider. A path π is d-separated (blocked) by a set of vertices Z if (1) π contains a chain $V_{i}\to V_{j}\to V_{k}$ or a fork $V_{i}\leftarrow V_{j}\to V_{k}$ with $V_{j}\in Z$; (2) π contains a v-structure $V_{i}\to V_{j}\leftarrow V_{k}$ with $De(V_{j},G)\notin Z$, and is d-connected otherwise [20]. Sets of vertices X and Y are d-separated by Z if and only if Z blocks all paths from any vertex $V_{i}\in X$ to any vertex $V_{j}\in Y$, denoted by $X⊥\;\;\;{⊥}_{G}Y|Z$. Furthermore, for any distribution *P*, $X⊥\;\;\;{⊥}_{P}Y|Z$ denotes that X and Y are conditional independent given Z. Given a DAG G and a distribution *P*, the Markov condition holds if $X⊥\;\;\;{⊥}_{P}Y|Z\Rightarrow X⊥\;\;\;{⊥}_{G}Y|Z$, while faithfulness holds if $X⊥\;\;\;{⊥}_{G}Y|Z\Rightarrow X⊥\;\;\;{⊥}_{P}Y|Z$. In fact, for any distribution, there exists at least one DAG such that the Markov condition holds, but there are some certain distributions that do not satisfy faithfulness to any DAG. Therefore, unlike the Markov condition, faithfulness is often regarded as an assumption. In this paper, unless otherwise stated, we assume that faithfulness holds, that is, $X⊥\;\;\;{⊥}_{G}Y|Z\Leftrightarrow X⊥\;\;\;{⊥}_{P}Y|Z$. For simplicity, we use the symbol ⊥⊥ to denote both (conditional) independence and d-separation.

From the concepts described, it can be inferred that a DAG characterizes the (conditional) independence relationships among a set of variables. In fact, multiple different DAGs may characterize the same conditional independent relationship. According to the Markov condition and faithfulness assumption, if the d-separation relationship contained in two DAGs is exactly the same, then these two DAGs are said to be Markov equivalent. Furthermore, two DAGs are Markov equivalent if and only if they share the same skeleton and v-structures [21]. All Markov equivalent DAGs constitute a Markov equivalent class, which can be represented by a completely partially directed acyclic graph (CPDAG) $G^{*}$. Two vertices are adjacent in the CPDAG $G^{*}$ if and only if they are adjacent in all DAGs in the equivalent class. The directed edge $V_{i}\to V_{j}$ in CPDAG $G^{*}$ indicates that this directed edge appears in all DAGs within the equivalent class, whereas the undirected edge $V_{i}-V_{j}$ signifies that $V_{i}\to V_{j}$ is present in some DAGs and $V_{i}\leftarrow V_{j}$ in others within the equivalent class [22]. A CPDAG is a chain graph [23] and can be learned by observational data and Meek’s rules [24] (Figure 2).

## 3. The Causal Graphical Model of Potential Factors and Functional Dynamic Target

Let $X=\{X_{1},\cdots ,X_{p}\}$ be a set of random variables representing potential factors and $Y=(Y_{1},\cdots ,Y_{q})$ be a functional dynamic target, where $Y_{i}$, for i=1,…,q, represents the state of the target at *q* different time points. Let G be a DAG defined over X∪Y, and let $G_{X}$ be the subgraph induced by G over the set of potential factors X. Suppose that the causal network of X can be represented by $G_{X}$, and when combined with the joint probabilities over X, denoted by P(·), we obtain a causal graphical model $\left(G_{X},P\right)$. Consequently, the data generation mechanisms of X and Y follow a causal Bayesian network model of $G_{X}$ and a model determined by the direct causes Pa(Y,G) of Y, respectively. Formally, we define a causal graphical model of the functional dynamic target as follows.

**Definition 1.** 
*Let G be a DAG over X∪Y, Pa(Y,G) denote the direct causes of Y in G and Ch(Y,G)=∅, P(·) be a joint distribution over X, and Θ be parameters determining the expectations of the functional dynamic target Y, which is influenced by Pa(Y,G). Then, the triple (G,P(·),Θ) constitutes a causal graphical model for Y if the following two conditions hold:*

*The pair $\left(G_{X},P\right)$ constitutes a Bayesian network model for X.*

*The functional dynamic target Y follows the following model:*

(1)
$$Y=\tilde{\mu }\left(\Theta \right)+{\tilde{\epsilon }}_{Y},$$

*where $\tilde{\mu }\left(\Theta \right)=\left(\mu \left(t_{1},\Theta \right),\cdots ,\mu \left(t_{q},\Theta \right)\right)$ is the vector of the mean function at time $t_{1},\ldots ,t_{q}$, and ${\tilde{\epsilon }}_{Y}=\left(\epsilon_{Y,t_{1}},\ldots ,\epsilon_{Y,t_{q}}\right)$ is the vector of error terms with mean of zero, that is, $E(\epsilon_{Y,t_{i}})=0,$i=1,…,q.*



Different functional dynamic targets use different mean functions. For example, the optimal mean function of growth curves of different species varies from the Gompertz function, $\mu \left(t,\left(a,b,c\right)\right)=ae^{-be^{-ct}}$, the Richards function, $\mu \left(t,\left(a,b,c,d\right)\right)=a/{\left(1+be^{-ct}\right)}^{d}$, the Hossfeld function, $\mu \left(t,\left(a,k,c\right)\right)=at^{k}/\left(c+t^{k}\right)$, and the Logistic function, $\mu \left(t,\left(a,r,c\right)\right)=a/\left(1+e^{-r(t-c)}\right)$ to Double-Logistic function, $\mu \left(t,\left(a_{1},r_{1},c_{1},a_{2},r_{2},c_{2}\right)\right)=a_{1}/\left(1+e^{-r_{1}\left(t-c_{1}\right)}\right)+a_{2}/\left(1+e^{-r_{2}\left(t-c_{2}\right)}\right)$ [25,26,27].

A causal graphical model of the functional dynamic target can be interpreted as a data generation mechanism of variables in X and Y as follows. First, the root variables in $G_{X}$ are generated according to their marginal probabilities. Then, following the topological ordering of the DAG $G_{X}$, for any non-root-variable *X*, when its parent nodes Pa(X,G) have been generated, *X* can be drawn from P(X∣Pa(X,G)), which is the conditional probability of *X* given its parent set Pa(X,G). Finally, the target is generated by Equation (1). According to Definition 1, the Markov condition holds for the causal graphical model of a dynamic target, that is, for any pair of variables $X_{i}$ and $X_{j}$, the d-separation of $X_{i}$ and $X_{j}$ given a set Z in G implies that $X_{i}$ and $X_{j}$ are conditionally independent given Z.

Given a mean function μ(t,Θ), we can estimate parameters $\hat{\Theta }$ as follows,
$$\hat{\Theta }=\arg\min_{\Theta }\sum_{i=1}^{n}\sum_{j=1}^{q}{\left(y_{i,t_{j}}-\mu \left(t_{j},\Theta \right)\right)}^{2},$$
where *n* and *q* represent the number of individuals and the length of the functional dynamic target, respectively. The residual sum of squares (RSS) is minimized at $\hat{\Theta }$. The Akaike information criterion (AIC) can be used to select the appropriate mean function $\mu^{*}$ to fit the functional dynamic targets. We have
$$\mu^{*}=\arg\min_{\mu }\left(nq+nq\log\left(2\pi \right)+nq\log\left(RSS/q\right)+2\right|\hat{\Theta }\left|\right).$$

## 4. Variable Screening for Causal Discovery

For the set of potential factors X and the functional dynamic target Y, our task is to find the direct causes and all causes of Y up to a given distance. An intuitive method involves learning the causal graph G to find all causes of Y. Alternatively, we could first learn the causal graph $G_{X}$ and then identify all variables that have directed paths to Y. However, as mentioned in Section 1, this intuitive approach has three main drawbacks. To address these challenges, we propose a variable screening method to reduce the number of potential factors, and a hypothesis testing method to test for (conditional) independence between potential factors and Y. By integrating these methods with structural learning approaches, we have developed an algorithm capable of learning and identifying all causes of functional dynamic targets.

Let *X* be a variable with level *K*. The variable *X* is not independent of Y if there exists at least two values of *X*, say $X=x_{1}$ and $X=x_{2}$, such that the conditional distributions of Y given $X=x_{1}$ and $X=x_{2}$ are different. Conversely, if the conditional distribution of Y given X=x remains unchanged for any *x*, we have that *X* and Y are independent. Let $\Theta_{x}$ be the parameter of the mean function of the functional dynamic target with X=x. To ascertain whether the variable *X* is not independent of Y, we implement the following test: (2)$$H_{0}:\;\Theta_{x}=\Theta_{x^{\prime }},\;\forall x,x^{\prime }\in \left\{1,\ldots ,K\right\},$$(3)$$H_{1}:\;\Theta_{x}\neq \Theta_{x^{\prime }},\;\exists x,x^{\prime }\in \left\{1,\ldots ,K\right\}.$$

Let $y_{i}=\left(y_{i,t_{1}},\ldots ,y_{i,t_{q}}\right)$ be the *i*th sample of the functional dynamic target with $X=x_{i}$. Under the null hypothesis, $y_{i}$ is modeled as $y_{i}=\tilde{\mu }\left(\Theta \right)+\tilde{\epsilon }$, whereas under the alternative hypothesis, it is modeled as $y_{i}=\tilde{\mu }\left(\Theta_{x_{i}}\right)+\tilde{\epsilon }$. Let lnL(Y,Θ) denote the unrestricted log-likelihood of Y under $H_{0}$ and let $\ln L\left(Y,\Theta^{H_{1}}\right)=\sum_{x=1}^{K}\ln L\left(Y,\Theta_{x}\right)$ denote the restricted log-likelihood of Y under $H_{1}$. The likelihood ratio statistic is calculated as follows:(4)$$LR=-2(\ln L\left(Y,\Theta \right)-\ln L\left(Y,\Theta^{H_{1}}\right)).$$

Under certain regular conditions, the statistic LR approximately follows $\chi^{2}$ distribution, and the degrees of freedom of this $\chi^{2}$ distribution are determined by the difference in the numbers of parameters between $H_{0}$ and $H_{1}$, as specified in Equations (2) and (3).

Therefore, by applying hypothesis tests described in Equations (2) and (3) to each potential factor, we can identify all variables significantly associated with the dynamic target. We denote these significant variables as $X_{sig}$, defined as $X_{sig}=\left\{X|X\in X,X\;\;/\;\;\;\;\;\;⊥\;\;\;⊥Y\right\}$. Indeed, since the mean function of the dynamic target depends on its direct causes, which in turn depend on indirect causes, the dynamic target ultimately depends on all its causes. Therefore, when *X* is precisely a cause of Y, we can reject the null hypothesis in Equation (2), implying that $X_{sig}$ includes all causes of the dynamic target, assuming no statistical errors. Therefore, given a dynamic target Y, perform hypothesis testing of $H_{0}$ against $H_{1}$ as defined in Equations (2) and (3) to each potential factor sequentially, then we can obtain the set $X_{sig}$ and their corresponding *p*-values ${\left\{X_{pv}\right\}}_{X\in X_{sig}}$, in which $X_{pv}$ is the *p*-value of the variable $X\in X_{sig}$.

A causal graphical model, as described in Definition 1, necessitates adherence to the Markov conditions for variables and the functional dynamic target. Given the Markov condition and the faithfulness assumption, a natural approach to identifying the causes of the functional dynamic target involves learning the causal structure of $X_{sig}$ and subsequently discerning the relationship between each variable $X\in X_{sig}$ and Y. For significant variables $X_{sig}$, we present the following theorem, with its proof available in Section A.2:

**Theorem 1.** 
*Suppose that (G,P,Θ) constitutes a causal graphical model for the functional dynamic target Y as defined in Definition 1, with the faithfulness assumption being satisfied. Let $X_{sig}$ denote the set comprising all variables in X that are dependent on Y. Then, the following assertions hold:*
*1.* 
*$X_{sig}$ consists of all causes and the descendants of these causes of Y, that is, $X_{sig}=An\left(Y,G\right)\cup De\left(An\left(Y,G\right),G\right)$.*
*2.* 
*For any two variables $X_{1},X_{2}\in X_{sig}$, if either $X_{1}$ or $X_{2}$ is a cause of Y, then $X_{1},X_{2}$ are not adjacent in $G_{X_{sig}}$ if and only if there exists a set $A\in X_{sig}$ such that $X_{1}⊥\;\;\;⊥X_{2}|A$.*
*3.* 
*For any two variables $X_{1},X_{2}\in X_{sig}$, if there exists a set $A\in X_{sig}$ such that $X_{1}⊥\;\;\;⊥X_{2}|A$, then $X_{1},X_{2}$ are not adjacent in $G_{X_{sig}}$.*



The first result of Theorem 1 implies the soundness and rationality of the method for finding $X_{sig}$ mentioned above. The second result indicates that when at least one end of an edge is a cause of Y, this edge can be accurately identified (in terms of its skeleton, not its direction) using any well-known structural learning methods, such as the PC algorithm [28] and GES algorithm [29]. Contrasting with the second result, the third specifies that for any pair of variables $X_{1},X_{2}\in X_{sig}$, if a separation set exists in $X_{sig}$ that blocks $X_{1},X_{2}$, then these variables are not adjacent in the true graph G. However, the converse does not necessarily hold due to the potential presence of a confounder or common cause $X_{3}\notin X_{sig}$, which can led to the appearance of an extraneous edge between $X_{1}$ and $X_{2}$ in the causal graph G derived solely from data on $X_{sig}$. To accommodate this, the CPDAG learned from $X_{sig}$ is denoted as $G_{X_{sig}}^{\prime }$, and the induced subgraph that corresponds to the true graph G over $X_{sig}$ is represented as $G_{X_{sig}}^{*}$. An illustrative example follows to elaborate on this explanation.

**Example 1.** 
*In Figure 3, Figure 3a presents a true graph defined over $X=\{X_{1},\ldots ,X_{5}\}$ and Y. Here, the set of significant variables is $X_{sig}=\left\{X_{1},X_{2},X_{3},X_{5}\right\}$, and $X_{4}$ is independent of Y. Figure 3b illustrates the induced subgraph $G_{1,X_{sig}}^{*}$ of the CPDAG $G_{1}^{*}$ over the set $X_{sig}$, while Figure 3c displays the graph learned through the structural learning method, such as the PC algorithm, applied to $X_{sig}$. It should be noted that, in $G_{1}$, $\left\{X_{4}\right\}$ is a separation set of $X_{1}$ and $X_{2}$, that is, $X_{1}⊥\;\;\;⊥X_{2}|X_{4}$. However, since $X_{4}\notin X_{sig}$ and structural learning only utilize data concerning $X_{sig}$, no separation set exists for $X_{1}$ and $X_{2}$ in $X_{sig}$. Consequently, $X_{1}$ and $X_{2}$ appear adjacent in the learned graph $G_{1,X_{sig}}^{\prime }$. Furthermore, given $X_{2}⊥\;\;\;⊥X_{3}$ and $X_{2}\;\;/\;\;\;\;\;\;⊥\;\;\;⊥X_{3}|X_{1}$, the structural learning method identifies a v-structure $X_{2}\to X_{1}\leftarrow X_{3}$. A similar process yiel $X_{1}\to X_{2}\leftarrow X_{5}$. Therefore, a bidirected edge $X_{1}↔X_{2}$ appears in the learned graph $G_{1,X_{sig}}^{\prime }$ but not in $G_{1,X_{sig}}^{*}$, as highlighted by the red edge in Figure 3c.*

*Similarly, Figure 3d presents a true graph $G_{2}$ defined over $X=\{X_{1},\ldots ,X_{5}\}$ and Y. In this scenario, the set of significant variables is identified as $X_{sig}=\left\{X_{1},X_{2},X_{3},X_{5}\right\}$, with $X_{4}$ being independent of Y. Figure 3b depicts the induced subgraph $G_{2,X_{sig}}^{*}$ of the CPDAG $G_{2}^{*}$ over $X_{sig}$, while Figure 3c illustrates the graph learned through the structural learning method, such as the PC algorithm, applied to $X_{sig}$. In $G_{2}$, the set $\left\{X_{1},X_{4}\right\}$ acts as a separation set between $X_{2}$ and $X_{3}$, indicating $X_{2}⊥\;\;\;⊥X_{3}|\left(X_{1},X_{4}\right)$. However, with $X_{4}\notin X_{sig}$ and structural learning relying solely on data concerning $X_{sig}$, a separation set for $X_{2}$ and $X_{3}$ in $X_{sig}$ no longer exists. As a result, $X_{2}$ and $X_{3}$ appear adjacent in the learned graph $G_{2,X_{sig}}^{\prime }$. Furthermore, given $X_{3}⊥\;\;\;⊥X_{5}$ and $X_{3}\;\;/\;\;\;\;\;\;⊥\;\;\;⊥X_{5}|X_{2}$, the structural learning method is capable of identifying a v-structure $X_{3}\to X_{2}\leftarrow X_{5}$. Therefore, a directed edge $X_{3}\to X_{2}$ is present in the learned graph $G_{2,X_{sig}}^{\prime }$ but not in $G_{2,X_{sig}}^{*}$, as highlighted by the red edge in Figure 3f.*


Example 1 illustrates two scenarios in which the graph $G_{X_{sig}}^{\prime }$ might include false positive edges that do not exist in $G_{X_{sig}}^{*}$. Importantly, these additional false edges may not appear between the causes of Y. Instead, they may occur between the causes and noncauses of Y, or exclusively among the noncauses of Y, as delineated in Theorem 1. The complete result is given by Proposition A1 in Section A.2. Indeed, a more profound inference can be drawn: the presence of extra edges does not compromise the structural integrity concerning the causes of Y, affecting neither the skeleton nor the orientation.

**Theorem 2.** 
*The edges in $E_{s}\left(G_{X_{sig}}^{\prime }\right)\setminus E_{s}\left(G_{X_{sig}}^{*}\right)$, if exists, do not affect the skeleton or orientation of edges among the ancestors of Y in $G^{*}$. Furthermore, we have $An\left(Y,G_{X_{sig}\cup Y}^{*}\right)=An\left(Y,G_{X_{sig}\cup Y}^{\prime }\right)$ and $PossAn\left(Y,G_{X_{sig}\cup Y}^{*}\right)\subseteq PossAn\left(Y,G_{X_{sig}\cup Y}^{\prime }\right)$, where $G_{X_{sig}\cup Y}^{*}$ and $G_{X_{sig}\cup Y}^{\prime }$ are graphs by adding a node Y and directed edges from Y’s direct causes to Y in graphs $G_{X_{sig}}^{*}$ and $G_{X_{sig}}^{\prime }$, respectively.*


According to Theorem 2, it is evident that although the graph $G_{X_{sig}}^{\prime }$ obtained through structural learning does not exactly match the induced subgraph of CPDAG $G^{*}$ over $X_{sig}$ corresponding to the true graph, the causes of the functional dynamic target Y in these two graphs are identical, including the structure among these causes. Thus, in terms of identifying the causes, the two graphs can be considered equivalent. Furthermore, Theorem 2 indicates that all possible ancestors of Y in $G_{X_{sig}\cup Y}^{*}$ are also possible ancestors in $G_{X_{sig}\cup Y}^{\prime }$, though the converse may not hold. The detailed proof is available in Section A.2.

**Example 2.** 
*The true graph G is given by Figure 4a, and the corresponding CPDAG $G^{*}$ is itself, that is, $G=G^{*}$. In this case, the set of significant variables is $X_{sig}=\left\{X_{1},X_{2},X_{4}\right\}$. Figure 4b is the induced graph $G_{X_{sig}\cup Y}^{*}$ of G ($G^{*}$) over $X_{sig}$, and Figure 4c is the CPDAG $G_{X_{sig}\cup Y}^{\prime }$ obtained by using the structural learning method on $X_{sig}$. Then, we have $An\left(Y,G_{X_{sig}\cup Y}^{*}\right)=An\left(Y,G_{X_{sig}\cup Y}^{\prime }\right)=\left\{X_{1}\right\}$, while $PossAn\left(Y,G_{X_{sig}\cup Y}^{*}\right)=\emptyset \subseteq \left\{X_{2},X_{4}\right\}=PossAn\left(Y,G_{X_{sig}\cup Y}^{\prime }\right)$.*


According to the causal graphical model in Definition 1 and the faithfulness assumption, Y is the sum of the mean function and an independent noise, and the mean function is a deterministic function of Y’s direct causes. Therefore, for any nondescendant of Y, say *X*, given the direct causes of Y, that is, Pa(Y,G), *X* is independent of Y. On the contrary, for any $X\in X_{sig}$, *X* is a direct cause of Y if and only if there is no subset A⊆Adj(X,G) such that X⊥⊥Y∣A.

Let A be a subset of $X_{sig}$. For any $X\in X_{sig}$ and X∉A, to test the conditional independence X⊥⊥Y∣A, consider the following test: (5)$$H_{0}:\;\Theta_{A=a,X=x}=\Theta_{A=a,X=x^{\prime }},\;\forall a,x,x^{\prime },$$(6)$$H_{1}:\;\Theta_{A=a,X=x}\neq \Theta_{A=a,X=x^{\prime }},\;\exists a,x,x^{\prime }.$$

Under the null hypothesis, the parameter only depends on the value of the set A, which can be denoted as $\Theta_{A}$, while under the alternative hypothesis, the parameter is determined by the values of both A and *X*, which can be denoted as $\Theta_{A,X}$. Let $\ln L(Y,\Theta_{A})$ be the log-likelihood of Y under $H_{0}$, and $\ln L(Y,\Theta_{A,X})$ be the log-likelihood of Y under $H_{1}$. The likelihood ratio statistic is
(7)$$LR=-2(\ln L\left(Y,\Theta_{A}\right)-\ln L\left(Y,\Theta_{A,X}\right)).$$

Under certain regular conditions, the statistic LR approximately follows $\chi^{2}$ distribution, with degrees of freedom equal to $|\Theta_{A,X}\left|-\right|\Theta_{A}|$.

Based on the above results, we propose a screening and structural learning-based algorithm to identify the causes of the functional dynamic target Y, as detailed in Algorithm 1.

In Algorithm 1, the initial step involves learning the structure over $X_{sig}$ utilizing data related to $X_{sig}$ through a structural learning method, detailed in Lines 1–6. The notation X∗−Y in Lines 3–4 signifies that the connection between *X* and *Y* could be either X−Y or X←Y. We first learn the skeleton of $X_{sig}$ following the same procedure as the PC algorithm (Line 1), with the details in Section A.1. Nevertheless, due to the potential occurrence of bidirected edges, adjustments are made in identifying v-structures (Lines 2–5), culminating in the elimination of all bidirected edges. According to Theorem 1, these bidirected edges, which are removed directly (Line 5), are present only between causative and noncausative variables or among noncausative variables of the functional dynamic target. Since these variable pairs are inherently (conditional) independent, removing such edges does not compromise the (conditional) independence relationships among the remaining variables, as shown in Theorem 2 and Example 1. Subsequently, we designate the set of direct causes as $DC:=X_{sig}$ and sequence these variables in ascending order of their correlations with Y (Lines 7–8). This is because variables with weaker correlation are less likely to be the direct cause of Y. Placing these variables at the beginning of the sequence can quickly exclude non-direct-cause variables in the subsequent conditional independence tests, thereby enhancing the algorithm’s efficiency, simplifying its complexity, and reducing the required number of conditional independence tests. Next, we add directed edges from all vertices in $X_{sig}$ to Y (Line 9) to construct the graph $G_{X_{sig}\cup Y}^{\prime }$. For each directed edge, say X→Y, we check the conditional independence of *X* and Y given a subset $A_{X}$ of DC (Lines 12–14). In seeking the separation set $A_{X}$, the search starts with single-element sets, progressing to sets comprising two elements, and so forth. Upon identifying a separation set, both vertices and directed edges are removed from DC and $G_{X_{sig}\cup Y}^{\prime }$, respectively (Lines 15–17). Lastly, if the separation set’s size *k* surpasses that of DC, implying that no conditional independence of *X* and Y can be found given any subset of DC, the directed edge X→Y remains in $G_{X_{sig}\cup Y}^{\prime }$.
**Algorithm 1** SSL: Screening and structural learning-based algorithm**Require:** $X_{sig}$ and their corresponding *p*-values ${\left\{X_{pv}\right\}}_{X\in X_{sig}}$, data sets about $X_{sig}$ and Y.
**Ensure:** Causes of Y.
 1:Learn the skeleton $G_{s}^{\prime }\left(X_{sig}\right)$ of the CPDAG $G_{X_{sig}}^{\prime }$ defined on $X_{sig}$ and obtain corresponding separation sets S based on the data set related to $X_{sig}$ via Algorithm A1 in Section A.1; 2:**repeat** 3:   Find the structure X∗−Y−∗Z satisfying X∉Adj(Z,G) in graph $G_{s}^{\prime }\left(X_{sig}\right)$; 4:   If Y∉S(X,Z), then orient as X∗→Y←∗Z; 5:**until** All structures X∗−Y−∗Z with X∉Adj(Z,G) in $G_{s}^{\prime }\left(X_{sig}\right)$ have been tested; 6:Construct the CPDAG $G_{X_{sig}}^{\prime }$ by deleting all bidirected edges and using Meek’s rules to orient as many undirected edges as possible in graph $G_{s}^{\prime }\left(X_{sig}\right)$; 7:Let $DC:=X_{sig}$; 8:Sort DC in ascending order of associations with Y using ${\left\{X_{pv}\right\}}_{X\in X_{sig}}$; 9:Let $G_{X_{sig}\cup Y}^{\prime }$ be the graph by adding a node Y to the graph $G_{X_{sig}}^{\prime }$,and for each $X\in X_{sig}$, add a directed edge X→Y to the graph $G_{X_{sig}\cup Y}^{\prime }$;10:Set k:=1;11:**while** k<|DC|, **do**12:   **for** each vertex X∈DC, **do**13:     **for** each subset $A_{X}$ of DC∖{X} with *k* vertices, **do**14:        Test the conditional independence $Y⊥\;\;\;⊥X|A_{X}$ using Equations (5) and (6);15:        **if** $Y⊥\;\;\;⊥X|A_{X}$, **then**16:           Delete the directed edge X→Y in graph $G_{X_{sig}\cup Y}^{\prime }$;17:           Let DC:=DC∖{X};18:        **end if**19:     **end for**20:   **end for**21:   k:=k+1;22:**end while**23:**return** $G_{X_{sig}\cup Y}^{\prime }$.

According to Theorem 1 and the discussion after Example 2, DC is the set of all direct causes of Y if all assumptions in Theorem 1 hold and all statistical tests are correct. Further, according to Theorem 2, all ancestors of Y can be obtained from the graph $G_{X_{sig}\cup Y}^{\prime }$. Therefore, Algorithm 1 can learn all the causes of Y correctly.

Note that in Algorithm 1, we first traverse the sizes of the separation set (Line 11) and then, for each given size, traverse all variables in the DC set and all possible separations with that size (Line 12 and 13) to test for the conditional independence of each variable and Y. That is, first fix the size of the separation set to 1, and then traverse all variables. After all variables are traversed once, increase the size of the separation set to 2, and then traverse all variables again. The advantage of this arrangement is that it can quickly remove the nondirect causes of Y and reduce the size of the DC set, thereby reducing the number of conditional independence tests and improving their accuracy. Furthermore, it is worth mentioning that the reason why we directly add directed edges from variables in DC to Y in graph $G_{X_{sig}\cup Y}^{\prime }$ (Line 9) is because we assume the descendant set of Y is empty, as shown in Definition 1, and in this case, Y’s adjacent set is exactly the direct causes we are looking for. If there is no such assumption, then it is necessary to judge the variables in Y’s adjacent set and distinguish the parents from the children.

## 5. A Screening-Based and Local Algorithm

Based on the previous results and discussions, we can conclude that Algorithm 1 is capable of correctly identifying the causes of a functional dynamic target. However, Algorithm 1 requires recovering the complete causal structure of $X_{sig}$ and Y. As analyzed in Section 1, learning the complete structure is unnecessary for identifying the causes of the target. Furthermore, Algorithm 1 may be influenced by the distance effect, whereby the correlation between a cause and the target may diminish from the data when the path from the cause to the target is too lengthy. Consequently, identifying this cause variable through observational data becomes challenging, potentially leading to missed causes. Therefore, we propose a screening-based and local approach to address these challenges.

In this section, we introduce a three-stage approach to learn the causes of functional dynamic targets. Initially, utilizing the causal graphical model, we apply a hypothesis testing method to screen variables, identifying factors significantly correlated with the target. Subsequently, we employ a constraint-based method to find the direct causes of the target from these significant variables. Lastly, we present a local learning method to discover the causes of these direct causes within any specified distance. We begin with the introduction of a screening-based algorithm that can learn the direct causes of Y, as shown in Algorithm 2.

In Algorithm 2, we initially set the set of direct causes $DC:=X_{sig}$ and arrange these variables in ascending order of their correlations with Y (Lines 1–2), which is the same as Algorithm 1. We introduce a set $N_{X}$ to contain variables determined not to belong to *X*’s separation set, starting as an empty set (Line 3). We then check the conditional independence of each variable X∈DC with Y. During the search for the separation set $A_{X}$, A is set as all subsets of $DC\setminus (X\cup N_{X})$ with *k* variables and is arranged roughly in descending order of their associations with Y (Lines 7–8). This is because variables that have a stronger correlation with Y are more likely to be the direct causes and are also more likely to become the separation set of other variables. Placing these variables at the beginning of the order can quickly find the separation set of nondirect causes and remove these variables from DC, which can reduce the number of conditional independence tests and accelerate the algorithm. Once we find the separation set $A_{X}$ for *X* and Y, we remove *X* from DC and add *X* to $N_{V}$ for each $V\in A_{X}$ (Lines 11–13). This is because when $A_{X}$ is the separation set of *X* and Y, the variables in $A_{X}$ appear in the path from *X* to Y. Consequently, *X* should not be in the separation set for variables in $A_{X}$ with respect to Y. Compared with Algorithm 1, introducing $N_{X}$ in Algorithm 2 improves efficiency and speed. While Algorithm 1 requires examining every subset of DC∖X (Line 8 in Algorithm 1), Algorithm 2 only needs to evaluate subsets of $DC\setminus (X\cup N_{X})$ (Line 7 in Algorithm 2). The theoretical validation of Algorithm 2’s correctness is presented below.
**Algorithm 2** Screening-based algorithm for learning direct causes of Y**Require:** $X_{sig}$ and their corresponding *p*-values ${\left\{X_{pv}\right\}}_{X\in X_{sig}}$, data sets about $X_{sig}$ and Y.
**Ensure:** Direct causes of Y.
 1:Let $DC:=X_{sig}$; 2:Sort DC in ascending order of associations with Y using ${\left\{X_{pv}\right\}}_{X\in X_{sig}}$; 3:Let $N_{X}:=\emptyset$ for each X∈DC; 4:Set k:=1; 5:**while** k<|DC|, **do** 6:   **for** each vertex *X* in DC, **do** 7:     Let A be the set of all subsets of $DC\setminus (\left\{X\right\}\cup N_{X})$ with *k* variables; 8:     Sort A approximately in descending order of associations with Y; 9:     **for** each $A_{X}\in A$, **do**10:        Test the conditional independence $Y⊥\;\;\;⊥X|A_{X}$ using Equations (5) and (6);11:        **if** $Y⊥\;\;\;⊥X|A_{X}$, **then**12:            Set DC:=DC∖{X}13:            Add *X* to $N_{V}$ for each $V\in A_{X}$;14:            break15:        **end if**16:     **end for**17:   **end for**18:   k:=k+119:**end while**20:**return** DC.


**Theorem 3.** 
*If all assumptions in Theorem 1 hold, and there are no errors in the independence tests, then Algorithm 2 can correctly identify all direct causes of Y.*


Next, we aim to identify all causes of Y within a specified distance. One natural method is to recursively apply Algorithm 2, starting with Y’s direct causes and then expanding to their direct causes. This process continues until all causes within the set distance are found. However, this method’s effectiveness for Y relies on the assumption that Y has no descendants, making its adjacent set its parent set. This is not the case for other variables. Thus, we must further analyze and distinguish variables in the adjacent set of other variables. Consequently, we introduce the LPC algorithm in Algorithm 3.
**Algorithm 3** LPC(T,U) algorithm**Require:** a target node *T*, a data set over variables X, a non-PC set U.**Ensure:** the PCD set of *T* and set S containing all separation relations.
 1:Set PCD:={X:X∈X∖U,andT  /⊥⊥X}; k:=1; S:=∅; 2:**while** k<|PCD|, **do** 3:   **for** each vertex X∈PCD, **do** 4:       **if** there exists $A_{X}\subseteq PCD\setminus \left\{X\right\}$ such that $|A_{X}|=k$ and $\left(T⊥\;\;\;⊥X|A_{X}\right)$, **then** 5:          Set PCD:=PCD∖{X} and add tuple $\left(T,X,A_{X}\right)$ to S; 6:       **end if** 7:   **end for** 8:   k:=k+1; 9:**end while**10:**return** PCD and S.


Algorithm 3 aims to learn the local structure of a given target variable *T*, but in fact, the final PCD set includes *T*’s Parents, Children, and Descendants. This is because when verifying the conditional independence (Line 4), we remove some nonadjacent variables of *T* in advance (Line 1), resulting in some descendant variables being unable to find the corresponding separation set.

**Example 3.** 
*In Figure 5, let $T=X_{1},U=\emptyset$. Since $X_{1}⊥\;\;\;⊥X_{4}$, we initially have $PCD=\{X_{2},X_{3}\}$ (Line 1 in Algorithm 3). Note that there originally exists a conditional independent relationship $X_{1}⊥\;\;\;⊥X_{3}|\left(X_{2},X_{4}\right)$ in the graph, but since we remove the vertex $X_{4}$ in advance, there is no longer a separation set of $X_{1}$ and $X_{3}$ in the set of PCD. Therefore, $X_{3}$ cannot be removed from PCD further and the output $PCD_{X_{1}}=\left\{X_{2},X_{3}\right\}$, that is, $X_{3}$, which is a descendant of $X_{1}$ but not a child of $X_{1}$, is included in $PCD_{X_{1}}$.*


Example 3 illustrates that there may indeed be some nonchildren descendants of the target variable in the PCD set obtained by Algorithm 3. Below, we show that one can identify these non-child-descendant variables by repeatedly applying Algorithm 3. For example, in Example 3, the PCD set of $X_{1}$ is $PCD_{X_{1}}=\left\{X_{2},X_{3}\right\}$. Then, we can apply Algorithm 3 to $X_{3}$ and find that the PCD set of $X_{3}$ is $PCD_{X_{3}}=\left\{X_{2},X_{4}\right\}$. It can be seen that $X_{1}$ is not in $PCD_{X_{3}}$. Hence, we can conclude that $X_{3}$ is a nonchildren descendant of $X_{1}$; otherwise, $X_{1}$ must be in $PCD_{X_{3}}$. Through this method, we can delete the non-child-descendant variables from the PCD set, so that the PCD set only contains the parents and children of the target variable. Based on this idea, we propose a step-by-step algorithm to learn all causes of a functional dynamic target locally, as shown in Algorithm 4.
**Algorithm 4** PC-by-PC: Finding all causes of target T within a given distance**Require:** a target set T⊆V=X∪Y, data set over V, and the maximum distance *m*.**Ensure:** all causes of T with length up to *m*.
 1:Set $n:=1,n^{\prime }:=0,CanC:=T$; 2:Initial graph G with directed edges from each vertex in T to an auxiliary node *L*; 3:**repeat** 4:   Set $X=CanC_{n}$; 5:   Let $U=\{V:X\notin PC_{V},\forall V\in CanC_{1:n-1}\}$; 6:   Get the PCD set and the separation set $\left(PC_{X},S_{X}\right)=LPC\left(X,U\right)$; 7:   **for** each $V\in PC_{X}\cap CanC_{1:n-1}$ **do** 8:     **if** $X\in PC_{V}$ **then** 9:        Add an undirected edge X−V to graph G;10:     **else**11:        $PC_{X}:=PC_{X}\setminus \left\{V\right\},PC_{V}:=PC_{V}\setminus \left\{X\right\}$;12:     **end if**13:   **end for**14:   Update G by modifying structures like $V_{1}-X-V_{2}$, $X-V_{1}-V_{2}$ and $X-V_{1}\leftarrow V_{2}$ to $V_{1}\to X\leftarrow V_{2}$, $X\to V_{1}\leftarrow V_{2}$ and $X\to V_{1}\leftarrow V_{2}$ respectively, if the middle vertex is not in the separation set of the two end vertices;15:   **if** *X* is the last vertex of CanC **then**16:     Update G by orientating undirected edges as much as possible via Meek’s rule;17:     **for** each $V\in CanC_{n^{\prime }:n}$, **do**18:        Add $PC_{V}\setminus CanC$ to the end of CanC if |Path(V,L)|<m or the *m*-th edge close to *L* in Path(V,L) is undirected;19:     **end for**20:   **end if**21:   $n^{\prime }:=n,n:=n+1$;22:**until** *X* is the last vertex of CanC; 23:**return** G and S.


In Algorithm 4, $CanC_{n}$ represents the *n*-th variable in the set CanC, and $CanC_{1:n-1}$ represents the first to the (n−1)-th variable in the set CanC. Path(V,L) denotes the shortest path from *V* to *L* in graph G. There are many methods to learn the shortest path, such as the Dijkstra algorithm [30]. Algorithm 4 uses the mentioned symmetric validation method to remove descendants from the PCD set (Lines 7–13), and hence, we directly write the PCD set as the PC set (Line 6). When our task is to learn all causes of a functional dynamic target Y, the target set T as the algorithm input is all direct causes of Y, which can be obtained by Algorithm 2, and the auxiliary node *L* is exactly the functional dynamic target Y (Line 2). In fact, we can prove it theoretically, as shown below.

**Theorem 4.** 
*If the faithfulness assumption holds and all independence tests are correct, then Algorithm 4 can learn all causes of the input target set T within a given distance m correctly. Further, if all assumptions in Theorem 1 holds, T is the set of direct causes of the functional dynamic target Y, and the auxiliary node L in Algorithm 4 is Y, then Algorithm 4 can learn all causes of Y within a given distance (m+1) correctly.*


Note that the above algorithm gradually spreads outward from the direct causes of Y, and at each step, the newly added nodes are all in the PC set of previous nodes (Line 18), which only involves the local structure of all causes of Y, greatly improving the efficiency and accuracy of the algorithm. Moreover, Algorithm 4 identifies the shortest path between each cause variable and Y. When the *m*-th edge on one path from Y cannot be oriented, it only continues to expand from that path, instead of expanding all paths (Line 18 in Algorithm 4), which simplifies the algorithm and reduces the learning of redundant structures.

## 6. Experiments

In this section, we compare the effectiveness of different methods for learning the direct and all causes of a functional dynamic target through simulation experiments. As mentioned before, to our knowledge, existing structural learning algorithms lack the specificity needed to identify causes of functionally dynamic targets, so we only compare the methods we proposed, which are as follows:SSL algorithm: The screening and structural learning-based algorithm given in Algorithm 1, which can learn both direct and all causes of a dynamic target simultaneously;S-Local algorithm: First, use the screening-based algorithm given in Algorithm 2, which can learn direct causes of a functional dynamic target, and then use the PC-by-PC algorithm given in Algorithm 4, which can learn all causes of a functional dynamic target.

In fact, our proposed SSL algorithm integrates elements of the screening method with those of traditional constraint-based structural learning techniques, as depicted in Algorithm 1. In its initial phase, the SSL algorithm is a modified version of the PC algorithm, extending its capabilities to effectively handle bidirectional edges introduced by the screening process. This extension of the PC algorithm, tailored to address the causes of the dynamic target, positions the SSL algorithm as a strong candidate for a benchmark.

In this simulation experiment, we randomly generate a causal graph G consisting of a dynamic target Y and *p* = (15, 100, 1000, 10,000) potential factors. Additionally, we randomly select 1 to 2 variables from these potential factors to serve as direct causes for Y. The potential factors are all discrete with finite levels, while the functional dynamic target $Y=(Y_{1},\cdots ,Y_{24})$ is a continuous vector, and its mean function is a Double-Logistic function, that is,
$$Y_{t}=\mu_{t|Pa(Y,G)}+\epsilon_{t},\;t=1,\ldots ,24,$$
where
$$\mu_{t|Pa(Y,G)}=\frac{a_{1|Pa(Y,G)}}{1+\exp(-r_{1|Pa(Y,G)}\left(t-c_{1|Pa(Y,G)}\right))}+\frac{a_{2|Pa(Y,G)}}{1+\exp(-r_{2|Pa(Y,G)}\left(t-c_{2|Pa(Y,G)}\right))},$$
and $\epsilon_{t}=\epsilon_{t-1}+\varepsilon_{t},\varepsilon_{t}∼N\left(0,0.02^{2}\right)$. The Pa(Y,G) in the subscript of the above equations indicates that parameters are affected by the direct causes of Y. For each causal graph G, we randomly generate the corresponding causal mechanism, that is, the marginal and conditional distributions of potential factors and the functional dynamic target, and generate the simulation data from it. We use different sample sizes n=(50,100,200,500,1000) and repeat the experiment 100 times for each sample size. In addition, we adopt adaptive significance level values in the experiment, because as the number of potential factors increases, the strength of screening also increases. In other words, as the number of potential factors *p* increases, the significance level α of the (conditional) independence test decreases. For example, α is 0.05 when p=100, while α is 0.0005 when *p* = 10,000.

To evaluate the effectiveness of different methods, suppose $X_{l}$ is the set of learned direct causes of Y by algorithms, and $X_{d}$ is the set of true direct causes of Y in the generated graph. Then, let $TP=|X_{l}\cap X_{d}|$, $FP=|X_{l}\setminus X_{d}|$, $FN=|X_{d}\setminus X_{l}|$, and we have
$$recall=\frac{TP}{TP+FN},\;precision=\frac{TP}{TP+FP},\;accuracy=\frac{p-FP-FN}{p},$$
where *p* is the number of potential factors. It can be seen that the recall measures how much the algorithm has learned among all the true direct causes. Precision measures how much of the direct causes learned by the algorithm are correct. Accuracy measures the proportion of correct judgments on whether each variable is a direct cause or not. The evaluation indicators for learning all causes can also be defined similarly.

The experiment results are shown in Table 1, in which time represents the total time (in seconds) consumed by the algorithm, and rec,prec,acc represent the average value of recall, precision, and accuracy over 100 experiments, respectively. In addition, different subscripts represent different methods. DC and AC denote that algorithms learn direct causes and all causes, respectively.

In Table 1, since the SSL algorithm obtains direct and all causes simultaneously through complete structural learning, for the sake of fairness, we only count the total time for both algorithms. It can be seen that the time of the two algorithms is approximately linearly related to the number of potential factors *p*. Moreover, when *p* is fixed, the algorithm takes longer and longer as the sample size *n* increases. In fact, for SSL algorithms, most of the time is spent on learning the complete graph structure. Therefore, as *n* increases, the (conditional) independence test becomes more accurate, resulting in an increase in the size of set $X_{sig}$ and a larger graph to learn, which naturally increases the time required. For the S-Local algorithm, more than 99% of the time is spent on optimizing the log-likelihood function during the (conditional) independence test in the screening stage. As *n* increases, the optimization time becomes longer and the total time also increases accordingly. This also explains why the time of the S-Local algorithm increases linearly as the number of variables increases, since the number of independence tests required increases roughly linearly. In addition, it can be seen that in most cases, the S-Local algorithm takes less time than the SSL algorithm, especially when *p* is small. However, when *p* is large, the time used by the two algorithms is similar. This is mainly because in this experimental setting, the mechanism of the functional dynamic target Y is relatively complex, and its mean function is a Double-Logistic function with too many parameters, which requires much time for optimization. In fact, even if there is only one binary direct cause, the mean function will have 13 parameters. When the mechanism of the functional dynamic target is relatively simple, the time required for the S-Local algorithm will also be greatly reduced. Besides, it should be noted that more than 99% of the time, the S-Local algorithm is used to check the independence in the screening step, and in practice, this step can be performed in parallel, which will greatly reduce the time required.

When learning the direct cause, whether it is recall, precision, or accuracy, the results of the S-Local algorithm are much higher than those of the SSL algorithm, especially the value of precision. The precision values of the SSL algorithm are very small, mainly because the accuracy of learning the complete graph structure is relatively low, resulting in learning many non-direct-cause variables in the local structure of Y. Particularly when *p* is large, it is difficult to correctly recover the local structure of Y. What’s more, it should be noted that under the same sample size, when *p* is small, the values of recall, precision and accuracy obtained by S-Local algorithm are not as good as those obtained when *p* is large. For example, when p=15, n=50, we have $rec_{S-Local}=0.500$, $prec_{S-Local}=0.474$ and $acc_{S-Local}=0.929$, but when *p* = 10,000, *n* = 50, we have $rec_{S-Local}$ = 1.000, $prec_{S-Local}$ = 1.000 and $acc_{S-Local}$ = 1.000. The recall and accuracy values of the SSL algorithm also show similar results. This result does not violate our intuition, as we use adaptive significance levels in the experiment. When *p* is large, in order to increase the strength of screening and facilitate subsequent learning of all causes, we use a smaller significance level. Therefore, the algorithm is more rigorous in determining whether a variable is the direct cause of Y when learning direct causes, making it easier to exclude those non-direct-cause variables.

When learning all causes, the recall and accuracy values of the SSL algorithm and S-Local algorithm increase monotonically with respect to the sample size, and even in cases with many potential factors, both algorithms can achieve very good results. For example, when *p* = 10,000, the accuracy values of both algorithms are above 99.9%. Of course, overall, the results of the S-Local algorithm are significantly better than those of the SSL algorithm. However, it should be noted that the values of precision of the two algorithms show different trends. The precision value of the SSL algorithm increases monotonically with *n* when *p* is large, but the trend is not significant when *p* is small. This is because the SSL algorithm is affected by the distance effect, and as *n* gradually increases, (conditional) independence tests also become more accurate. As a result, many causes that are far away from Y can be identified. When *p* is large, the number of causes that are far away from Y is also large. Therefore, the precision of the SSL algorithm will gradually increase. However, when *p* is small, most variables have a short distance from Y. Although the SSL algorithm can also obtain more causes (the value of $rec_{SSL}$ increases), it also includes some noncause variables that are strongly related to Y in the set of causes. At this time, the value of precision does not have a clear trend. On the other hand, the precision value of the S-Local algorithm monotonically increases with respect to *n* when *p* is small, and as *p* gradually increases, this trend gradually transforms into a monotonic decrease. This is because when *p* is small, as *n* increases, the S-Local algorithm can identify more causes through a more accurate (conditional) independence test. However, when *p* is large, the number of noncause variables obtained by the S-Local algorithm is greater than the number of causes. Therefore, the recall value still increases, but the precision value gradually decreases. In other words, in this case, there is a trade-off between the values of recall and precision of the S-Local algorithm. However, it should be noted that although the trends of precision values are different, the accuracy values of both algorithms increase with the increase in sample size.

It should be noted that the primary objective of the models and algorithms introduced in this paper is to identify the causes of functional dynamic targets, addressing the "Cause of Effect" (CoE) challenge, rather than directly predicting Y. However, based on the causal graphical model for these targets, correctly identifying Y’s direct causes is indeed sufficient for making accurate predictions. In the simulation experiment, with 15 nodes and 1000 samples, the Mean Squared Error (MSE) of prediction is 0.281 for simulations that incorrectly learn Y’s direct causes. This figure dropped to 0.185 when the causes were correctly identified, reflecting a significant reduction in prediction error of approximately 34%. Additionally, as illustrated in Table 1, the S-Local algorithm demonstrated exceptional accuracy in identifying the direct causes, with a success rate consistently above 98% in most cases. This high level of accuracy indicates that our algorithms perform well in predicting Y as well.

## 7. Discussion and Conclusions

In this paper, we first establish a causal graphical model for functional dynamic targets and discuss hypothesis testing methods for testing the (conditional) independence between random variables and functional dynamic targets. In order to deal with situations where there are too many potential factors, we propose a screening algorithm to screen out some variables that are significantly related to the functional dynamic target from a large number of potential factors. On this basis, we propose the SSL algorithm and S-Local algorithm to learn the direct causes and all causes within a given distance of functional dynamic targets. The former utilizes the screening algorithm and structural learning methods to learn both the direct and all causes of functional dynamic targets simultaneously by recovering the complete graph structure of the screened variables. Its disadvantage is that learning the complete structure of the graph is very difficult and redundant, and it is also affected by the distance effect, resulting in a low accuracy in learning causes. The latter first uses a screening-based algorithm to learn the direct causes of functional dynamic targets, and then uses our proposed PC-by-PC algorithm, a step-by-step locally learning algorithm, to learn all causes within a given distance. The advantage of this algorithm is that all learning processes are controlled within the local structure of current nodes, making the algorithm no longer affected by the distance effect. In fact, this algorithm only focuses on the local structure of each cause variable, rather than learning the complete graph structure, greatly saving time and space. Moreover, the algorithm not only pays attention to the distance, but also can identify the direct path between each cause variable and the functional dynamic target, so that the algorithm does not need to identify the whole structure of a certain part but only learns the part of the local structure involving the cause variables, further reducing the learning of redundant structures.

It should be noted that when the causal mechanism of functional dynamic targets is very complex, the time required for the S-Local algorithm may greatly increase. In addition, the choice of significance level will also have an impact on the precision of the algorithm. Thus, how to simplify the causal model of functional dynamic targets and how to reasonably choose an appropriate significance level are two directions of our future work.

## Figures and Tables

**Figure 1 entropy-26-00541-f001:**
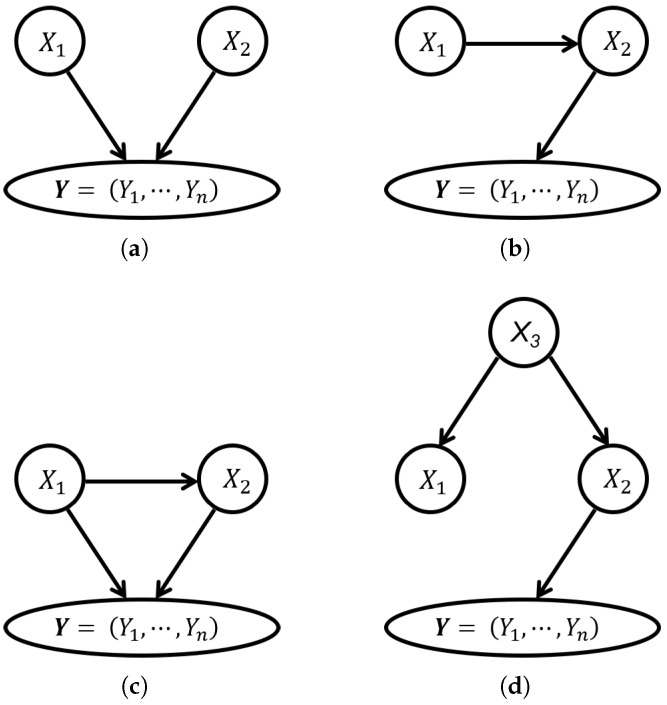
The causal graphs when the variable $X_{1}$ is significantly associated with Y: (**a**) V-structures. (**b**) Chains. (**c**) Triangles. (**d**) Forks.

**Figure 2 entropy-26-00541-f002:**
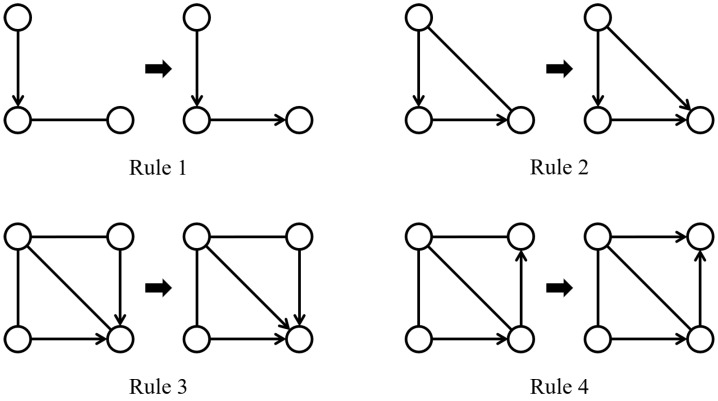
Meek’s rules comprise four orientation rules. If the graph on the left-hand side of a rule is an induced subgraph of a PDAG, then the corresponding rule can be applied to replace an undirected edge in the induced subgraph with a directed edge. This replacement results in the induced subgraph transforming into the graph depicted on the right-hand side of the rule.

**Figure 3 entropy-26-00541-f003:**
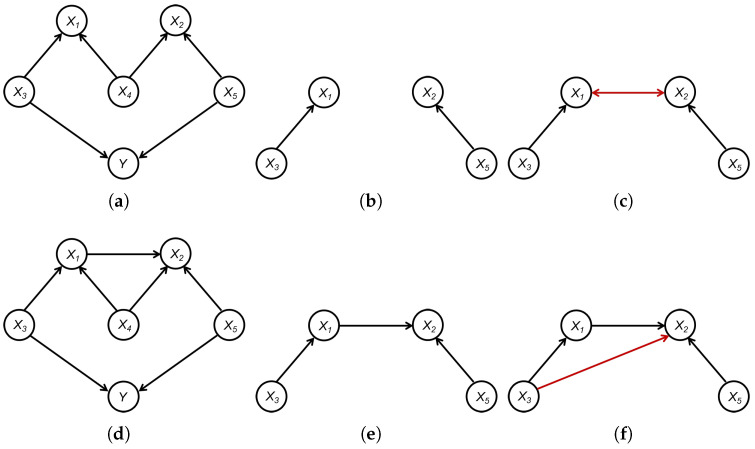
An example to illustrate the difference between $G_{X_{sig}}^{\prime }$ and $G_{X_{sig}}^{*}$: (**a**) True graph $G_{1}$. (**b**) $G_{1,X_{sig}}^{*}$. (**c**) $G_{1,X_{sig}}^{\prime }$. (**d**) True graph $G_{2}$. (**e**) $G_{2,X_{sig}}^{*}$. (**f**) $G_{2,X_{sig}}^{\prime }$.

**Figure 4 entropy-26-00541-f004:**
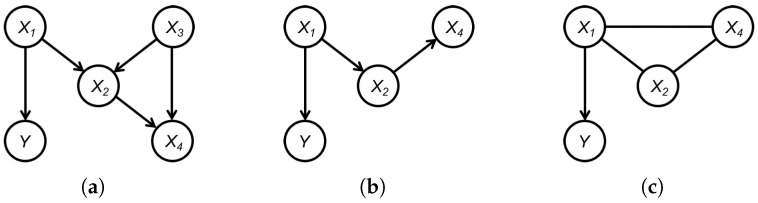
An example to illustrate the results in Theorem 2: (**a**) True graph G. (**b**) $G_{X_{sig}\cup Y}^{*}$. (**c**) $G_{X_{sig}\cup Y}^{\prime }$.

**Figure 5 entropy-26-00541-f005:**
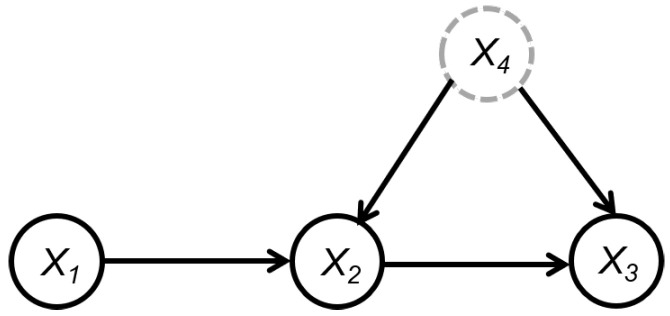
An example to illustrate the PCD set obtained by Algorithm 3.

**Table 1 entropy-26-00541-t001:** Experimental results of SSL algorithm and S-Local algorithm under different settings.

*p*	*n*	${\mathrm{time}}_{\mathrm{SSL}}$	${\mathrm{time}}_{S-\mathrm{Local}}$	Cause	${\mathrm{rec}}_{\mathrm{SSL}}$	${\mathrm{rec}}_{S-\mathrm{Local}}$	${\mathrm{prec}}_{\mathrm{SSL}}$	${\mathrm{prec}}_{S-\mathrm{Local}}$	${\mathrm{acc}}_{\mathrm{SSL}}$	${\mathrm{acc}}_{S-\mathrm{Local}}$
15	50	50	55	DC	0.487	0.500	0.352	0.474	0.866	0.929
				AC	0.157	0.487	0.721	0.814	0.445	0.640
	100	64	49	DC	0.557	0.829	0.485	0.814	0.905	0.975
				AC	0.143	0.656	0.786	0.877	0.451	0.733
	200	112	60	DC	0.538	0.885	0.386	0.856	0.888	0.981
				AC	0.224	0.882	0.603	0.878	0.476	0.854
	500	676	85	DC	0.551	0.939	0.466	0.929	0.927	0.989
				AC	0.363	0.977	0.567	0.893	0.552	0.916
	1000	1126	126	DC	0.778	0.981	0.691	0.963	0.957	0.996
				AC	0.566	0.996	0.702	0.890	0.667	0.923
100	50	251	277	DC	0.293	0.283	0.072	0.256	0.934	0.984
				AC	0.112	0.223	0.154	0.358	0.867	0.915
	100	224	227	DC	0.398	0.755	0.141	0.656	0.956	0.993
				AC	0.104	0.604	0.226	0.688	0.884	0.946
	200	290	235	DC	0.292	0.917	0.113	0.828	0.962	0.996
				AC	0.123	0.891	0.205	0.763	0.891	0.961
	500	890	336	DC	0.221	0.916	0.071	0.900	0.962	0.997
				AC	0.145	0.966	0.156	0.753	0.892	0.957
	1000	1509	527	DC	0.462	0.978	0.156	0.962	0.967	0.999
				AC	0.327	0.996	0.308	0.668	0.908	0.933
1000	50	836	839	DC	0.814	1.000	0.073	1.000	0.989	1.000
				AC	0.336	0.204	0.235	0.924	0.985	0.992
	100	936	962	DC	0.860	1.000	0.069	1.000	0.988	1.000
				AC	0.587	0.573	0.379	0.867	0.988	0.995
	200	1204	1222	DC	0.980	1.000	0.083	1.000	0.989	1.000
				AC	0.724	0.847	0.446	0.861	0.989	0.997
	500	2376	1930	DC	1.000	1.000	0.118	1.000	0.992	1.000
				AC	0.804	0.922	0.500	0.814	0.991	0.997
	1000	4015	3118	DC	1.000	1.000	0.121	1.000	0.993	1.000
				AC	0.873	0.998	0.523	0.813	0.992	0.998
10,000	50	9109	9480	DC	0.667	1.000	0.150	1.000	1.000	1.000
				AC	0.148	0.148	0.194	1.000	0.999	0.999
	100	10,008	10,463	DC	0.654	1.000	0.101	1.000	0.999	1.000
				AC	0.376	0.538	0.285	0.884	0.999	1.000
	200	13,710	13,836	DC	0.923	1.000	0.101	1.000	0.999	1.000
				AC	0.551	0.833	0.395	0.871	0.999	1.000
	500	21,084	18,343	DC	1.000	1.000	0.130	1.000	0.999	1.000
				AC	0.782	0.919	0.502	0.813	0.999	1.000
	1000	31,476	31,862	DC	1.000	1.000	0.126	1.000	0.999	1.000
				AC	0.813	0.959	0.505	0.787	0.999	1.000

## Data Availability

The simulated data can be regenerated using the codes, which can be provided to the interested user via an email request to the correspondence author.

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
