# Peer review of "Learning Causes of Functional Dynamic Targets: Screening and Local Methods"

_entropy, 2024, doi:10.3390/e26070541_

Round 1

Reviewer 1 Report

Comments and Suggestions for Authors

Comment on "Learning Causes of Functional Dynamic Targets: Screening and Local Methods"

The authors introduce a causal graphical model that integrates Bayesian networks with functional dynamic targets, effectively mapping out the causal relationships among variables and the target. They further introduce a dimension-reduction screening method that efficiently identifies the causes of a specific target, ensuring that all identifiable causes are accurately determined. The authors develop a screening-based, local approach to ascertain the causes influencing the functional dynamic target, regardless of the distance among various factors.

(1) The algorithm 1 in page 7 simply identifies the list of variables from X, which are significant. Do authors really need to call it as an algorithm? This information can be summarised in a single sentence.

(2) Screening the variable to identify the Xsig, each component is tested independently. Does this lead to omitted variable bias? Further, it could affect the statistical significance of the results.

(3) What is the rational to choose Double-Logit function for the simulation exercise? How stable are the results with the other data generating process?

(4) It would be nice to include an empirical example to illustrate some of the key findings.

Author Response

Thank you for your valuable comments and suggestions. Below, we provide a point-by point response to your  feedback.

Q(1) The algorithm 1 in page 7 simply identifies the list of variables from X, which are significant. Do authors really need to call it as an algorithm? This information can be summarized in a single sentence.

A(1) This is indeed a good suggestion. We make a modification and summarize the method in a single sentence following your suggestion.  

Q(2) Screening the variable to identify the Xsig, each component is tested independently. Does this lead to omitted variable bias? Further, it could affect the statistical significance of the results.

A(2) Screening the variables to identify the Xsig set, where each component is tested independently, does not inherently lead to omitted variable bias. This process is based on the assumption that the faithfulness condition and the Markov condition are satisfied, which theoretically ensures that all variables that have a direct or indirect causal relationship with the dependent variable Y can be included in the Xsig set.

However, in practical applications, hypothesis testing may encounter errors, which can be classified into two types:

Type I Error (False Positive): This occurs when an independent variable is incorrectly identified as being dependent on Y, leading to the inclusion of redundant variables in the model. 

Type II Error (False Negative): This is the error of concern, where a variable that is actually dependent on Y is mistakenly considered independent and thus omitted from the model.  

Despite these potential errors, the process of variable screening does not necessarily exacerbate the problem of omitted variables.  This is especially true if the causal path between the variable and Y is direct. Even if such variables are not screened out, their direct causal relationship with Y may not be revealed if they show statistical independence from Y. 

However, when the causal relationship is more complex, involving indirect paths, the risk of Type II error increases. This is where our proposed local algorithm comes into play.  In the local algorithm, we screen variables specifically to learn the direct causes of Y, without screening for indirect causes, thereby reducing the risk of omitted variable bias.

Q(3) What is the rational to choose Double-Logit function for the simulation exercise? How stable are the results with the other data generating process?

A(3) The Double-Logistic function is chosen for the simulation exercise due to its prevalence and versatility in modeling growth patterns. It is a common choice for representing S-shaped growth curves, which are typical in many biological and social phenomena. The Double-Logistic function is particularly notable for its complexity and adaptability, making it an excellent candidate for demonstrating the robustness of our algorithms.

In addition to its widespread use, the Double-Logistic function has more parameters, which makes it one of the most complex growth curve functions. For instance, the Double-Logistic function typically includes six main parameters: two for the asymptotes (upper and lower), two for the growth rates, and two for the inflection points. In contrast, the standard Logistic function has three main parameters, and the Power function has only two. This complexity allows the Double-Logistic function to capture a wide range of growth dynamics, including those with multiple phases or inflection points.

Choosing the Double-Logistic function for the simulation exercise also allows us to evaluate how well our algorithms can handle a complex scenarios. When other, simpler data generation processes are used, the algorithms typically require less computational time due to the reduced complexity and number of parameters.

It is important to note that while the Double-Logistic function is chosen for its complexity in the simulation, the choice of model in actual analysis should be based on the specific characteristics of the data and the research question at hand.

Q(4) It would be nice to include an empirical example to illustrate some of the key findings.

A(4) We agree with your perspective.  On the one hand, within the realm of accessible open data, we have not been able to find a suitable dataset that meets our requirements. On the other hand, for datasets mentioned in literature but not publicly available, we have encountered some challenges in obtaining the necessary permissions for their use.

Nevertheless, we believe that employing simulated data can still achieve the desired evaluative outcomes. With simulation experiments, we know the complete structure of the true causal graph. Therefore, we can compare the structure in the true causal graph with causes of Y we have learned, in order to measure the effectiveness and accuracy of our algorithms. However, for real data, we usually cannot know what the exact true causal graph looks like. In this case, to determine whether the causes we have learned are correct, we can only rely on the experts’ knowledge, which is not a very accurate and intuitive way of judgment.

Reviewer 2 Report

Comments and Suggestions for Authors

There are several aspects that need to be clarified and explained more clearly:

- The paper introduces and compares two approaches, this should be clarified since the introduction (mentioned in the conclusions)

- What is the definition of "distance"

- Do you assume no cofounders (causal sufficiency)?

- It seems you assume the factors are independent?

- As "Y" is a vector, how do you evaluate the dependence relations with the variables Xi, explain.

- Does the method depend on the order of the independence tests? 

There are some parts that seem incorrect:

Line 331 - non-ancestor independent given its direct causes - actually should be its non descendants 

Example 3 - seems wrong

Theorem 3 - no proof

Regarding the experiments:

- A comparison with some base line method is required, for instance PC

- Indicate how many direct causes are in the randomly generated causal graph

- Evaluate how good are the learned causal models for predicting "Y"

Author Response

Thank you for your valuable comments and suggestions. Below, we provide a point-by point response to your feedback.  

Q(1) The paper introduces and compares two approaches, this should be clarified since the introduction (mentioned in the conclusions)

A(1) We acknowledge the need for clarity regarding the introduction and comparison of the two approaches in our paper. To address this, we have revised the introduction section to ensure that the methodologies are clearly outlined and their distinctions are highlighted from the outset.

Q(2) What is the definition of "distance"

A(2) The distance between a cause variable and the dynamic target is the length of the shortest directed path from the cause variable to the dynamic target. This path represents the minimal number of steps or intermediary variables that need to be traversed to get from the cause to the target.

Q(3) Do you assume no cofounders (causal sufficiency)?

A(3) Indeed, our analysis is predicated on the assumption of causal sufficiency, which we explicitly acknowledge in our paper by stating "… assuming no hidden variables or selection biases."  

Q(4) As "Y" is a vector, how do you evaluate the dependence relations with the variables Xi, explain.

A(4) We propose the hypothesis tests to test the marginal independence between the potential factor and the dynamic target Y, as shown in Equations (2) and (3). Then, using the likelihood ratio statistic given in Equation (4), which approximately follows a chi-square distribution, we can evaluate the dependence relations between Xi and Y. The conditional independence test can be defined similarly, as shown in Equations (5) – (7).

Q(5) Does the method depend on the order of the independence tests?

A(5) Partially, yes, the method's outcome can be influenced by the order of the independence test.

Theoretically, if all independence tests are conducted without error, the sequence in which they are performed should not influence the outcome of the algorithm. However, in practical analysis, independence tests are subject to two types of errors. Regardless of the type of error, the initial structural errors determined by the first independence tests can be related to the choice of conditioning sets in subsequent tests and the application of the meek rules, which may affect the subsequent learning results. Therefore, the results of constraint-based  algorithms (say, the original PC algorithm) are often related to the order of testing.

Our proposed SSL algorithm is a hybrid of the screening method and conventional constraint-based structural learning algorithms (PCs). As such, it may also be influenced by the order of testing.

Furthermore, the local algorithm we introduced initiates the learning process from the outcome Y, first identifying direct causes and then learning causes at a distance of 2, progressively moving to more distant causes. This step-by-step approach with a more or less fixed sequence may result in a lower susceptibility to the influence of the order of independence tests.

Q(6) Line 331 - non-ancestor independent given its direct causes - actually should be its non descendants.

A(6) You are correct, it is indeed a typo.  Thank you for bringing this to our attention.  The corrected sentence should read: "The non-descendants are independent of Y given Y's direct causes."

Q(7) Example 3 - seems wrong

A(7) We appreciate the opportunity to review Example 3. This example is intended to illustrate that sometimes the output of the LPC (Least Parents Children) algorithm may include non-children descendants rather than just parents and children. It also explains that by repeatedly applying the LPC algorithm, one can identify these non-children descendant variables. In Figure 5, let T = X1, the initial PCD set for X1 is PCD = {X2, X3}. Since X1 is directly connected to X2, and X1 is not independent of X3 given X2, there is no separating set for X1 and X2, as well as for X1 and X3, within PCD. Consequnetly, the output PCDX1 is {X2, X3}, where X3 is not a child of X1. Let T = X3, and applying LPC again, we have that PCDX3 is {X2, X4}. It can be seen that X1 is not in PCDX3.  Therefore, we can conclude that X3 is a non-children descendant of X1, otherwise, X1 must be in PCDX3.

We have revised Example 3 to make it clearer. It reads:

Example 3. In Figure 5, let T = X1, U = . Since X1  X4, we initially have PCD = {X2, X3}  (Line 1 in Algorithm 3) Note that there originally exists a conditional independent relationship X1  X3 | (X2, X4) in the graph, but since we remove the vertex X4 in advance, there is no longer  a separation set of X1 and X3 in the set of PCD . Therefore, X3 cannot be removed from PCD further  and the output PCDX1 = {X2, X3}, that is, X3, which is a descendant of X1 but not a child of X1, is included in PCDX1 .

Example 3 illustrates that there may indeed be some non-children descendants of  the target variable in the PCD set obtained by Algorithm 3. Below, we show that one can  identify these non-child descendant variables by repeatedly applying the Algorithm 3. For example, in Example 3, the PCD set of X1 is PCDX1 = {X2, X3}. Then we can apply  Algorithm 3 to X3 and find that the PCD set of X3 is PCDX3 = {X2, X4}. It can be seen that X1 is not in PCDX3 . Hence, we can conclude that X3 is a non-children descendant of X1, otherwise, X1 must be in PCDX3 . Through this method, we can delete the non-children descendant variables from the PCD set, so that the PCD set only contains the parents and children of the target variable. Based on this idea, we propose a step-by-step algorithm to learn all causes of a functional dynamic target locally, as shown in Algorithm 4.

Q(8) Theorem 3 - no proof

A(8) We acknowledge the inquiry regarding the proof of Theorem 3. Upon review, we have verified that the proof was indeed included in the document, as indicated in lines 762 to 774 of the previous version, and now appears in lines 803 to 815 of the current version.

Q(9) A comparison with some base line method is required, for instance PC.

A(9) We appreciate the suggestion for a comparative analysis with a baseline method, such as the PC algorithm. We also discuss it in this version.

  "...  to our knowledge, existing structural learning algorithms lack the specificity needed to identify causes of functionally dynamic targets...

"In fact, our proposed SSL algorithm integrates elements of the screening method with those of traditional constraint-based structural learning techniques, as depicted in Algorithm~\ref{alg:SSL}. In its initial phase, the SSL algorithm is a modified version of the PC algorithm, extending its capabilities to effectively handle bidirectional edges introduced by the screening process. This extension of the PC algorithm, tailored to address the causes of the dynamic target, positions the SSL algorithm as a strong candidate for a benchmark."

Q(10) Indicate how many direct causes are in the randomly generated causal graph

A(10) Thank you for your attention to detail. In this version, we have specified that in our randomly generated causal graphs, the functional dynamic target has one to two direct causes.  It reads:  Additionally, we randomly select 1 to 2 variables from these potential factors to serve as direct causes for  Y.

Q(11) Evaluate how good are the learned causal models for predicting "Y"

A(11) We have summarized the results of experiments and added a paragraph to evaluate the prediction performance for Y.  It reads:

 “It should be noted that the primary objective of the models and algorithms introduced in this paper is to identify the causes of functional dynamic targets, addressing the 'Cause of Effect' (CoE) challenge, rather than directly predicting Y. However, based on the causal graphical model for these targets, correctly identifying Y's direct causes is indeed sufficient for making accurate predictions (or interventional predictions). In the simulation experiment, with 15 nodes and 1000 samples, the Mean Squared Error (MSE) of prediction is 0.281 for simulations that incorrectly learn Y's direct causes.  This figure dropped to 0.185 when the causes were correctly identified, reflecting a significant reduction in prediction error of approximately 34%. Additionally, as illustrated in Table 1, the S-Local algorithm demonstrated exceptional accuracy in identifying the direct causes, with a success rate consistently above 98% in most cases. This high level of accuracy indicates that our algorithms perform well in predicting Y as well.”